# Using honeybees for national scale long-term eDNA biomonitoring

**Jennifer M. G. Shelton**[1☾], **Ben A. Woodcock**[1☾]*, **Lindsay Newbold**[1☾], **Anna Oliver**[1],
**Joanna Savage**[1], **Eleanor Grove**[1], **Manasa Suresh**[1], **Ujala Syed**[1], **Lauren Cook**[1],
**Mike Brown**[2], **Biren Rathod**[1], **Michael Bennett**[1], **Jim Bacon**[2], **Emily V. Upcott**[1],
**Daniel S. Read**[1], **Katharine Turvey**[1], **David Roy**[1], **Hyun Gweon Soon**[1,3], **Richard Pywell**[1]

**1** UK Centre for Ecology & Hydrology, Wallingford, United Kingdom, **2** UK Centre for Ecology &
Hydrology, Lancaster, United Kingdom, **3** School of Biological Sciences, University of Reading, Reading,
United Kingdom

☾ These authors contributed equally to this work.
* bawood@Ceh.ac.uk

journal.pone.0347485

CANADA

**Peer Review History:** PLOS recognizes the
benefits of transparency in the peer review
process; therefore, we enable the publication
of all of the content of peer review and
author responses alongside final, published
articles. The editorial history of this article is
available here: https://doi.org/10.1371/journal.
pone.0347485

## Abstract

As central place foragers, bees integrate information over large spatial scales on diet
and pollutant exposure, offering insights into environmental impacts on their popula-
tions. Data from bee biomonitoring has strong applied and policy relevance, particu-
larly when conducted over extensive spatial and temporal scales. However, practical
challenges have limited large-scale sustainable implementation of such monitoring
networks beyond relatively small-scale experimental studies. This paper describes
the creation of a national, citizen science–led honeybee biomonitoring platform.
Citizen scientist beekeepers provide biological samples at a national scale that would
be cost prohibitive to replicate using conventional sampling strategies. Environmen-
tal DNA (eDNA) within honey allows quantification of spatial and temporal patterns
in foraging resources. From 2018–2025, over 3,500 beekeepers have contributed
5,789 honey samples from across England, Wales, Scotland, and Northern Ireland.
Most samples are collected between May and October and originate from intensively
managed agricultural land (54% land use cover), urban and suburban areas (25%),
forests (13%), and extensively managed landscapes (8%). eDNA analyses from
2018–2022 reveal strong temporal and spatial variation in plant resource use. Brassi-
cas (wild and crop species such as oilseed rape), clovers (Trifolium spp.), and bram-
bles (Rubus spp.) dominate honeybee diets, alongside notable use of invasive plants.
Large-scale, long-term monitoring of floral resource use by honeybees establishes a
benchmark for assessing resource availability to wider pollinator communities. The
scheme provides data to interpret land-use change, agri-environmental policy out-
comes, and climate-driven shifts in flowering resources. Archived honey samples also
support future research on invasive species, bee pathogens, and chemical (including
pesticide) exposure. The combination of citizen science and eDNA methods enables

**Data availability statement:** If published data associated with the NHMS monitoring scheme describing country level eDNA derived foraging patterns for the UK. These will be published on the NERC Environmental Informatics Data Centre with full DOI. Original eDNA sequence data underpinning this will also be published on the GenBank open access nucleotide sequence database. All data and code used to produce figures is available at GitHub at https://github.com/BenAWoodcock/NHMS-Scheme-overview-code.

**Funding:** This research was funded by the Natural Environment Research Council (NERC) and the Biotechnology and Biological Sciences Research Council (BBSRC) joint research programs NE/N018125/1 LTS-M ASSIST—Achieving Sustainable Agricultural Systems (www.assist.ceh.ac.uk) and NE/W005050/1 AgZero+: Towards sustainable, climate-neutral farming.

**Competing interests:** The authors have declared that no competing interests exist.

cost-effective, nationwide ecological monitoring at a scale unattainable through traditional approaches.

## Introduction

Insect pollinators provide critical ecosystem services worldwide, with an estimated contribution to UK crop yields of *c.* £500 million annually [1]. Wild insects providing pollination have undergone worldwide declines [2], with a 33% decline in wild bees and hoverflies in Great Britain [3]. The causes for declines are diverse, and include pesticides, diseases, loss of nesting sites, invasive species and climate change [2–5]. However, habitat loss and landscape simplification reducing access to critical floral foraging resources has been a major contributory factor that potentially interacts with other drivers [2,4,6]. Longer-term foraging patterns provide key insights into resource utilisation likely to impact on these population level processes [7,8]. Understanding such large-scale and long-term trends in the availability of these floral foraging resources has significant implications for understanding the impacts of land use change, climatic shifts in plant communities or temporal mismatch in resource [2,4,6,9].

Filling this knowledge gap at the national scales is a daunting task with significant practical problems beyond the resources of most research projects. Conventional monitoring using trained researchers is costly, requires taxonomic expertise and can normally only be implemented for a restricted number of sites with biological observation occurring for fixed time windows [10]. These issues are encountered in systematic monitoring of insect populations, for example the UK and European Pollinator Monitoring Schemes [10] or the US National Native Bee Monitoring Network [11]. While these schemes often capture foraging events, they are dependent on snapshot sampling that may fail to quantify the full spectrum of foraging interactions [12,13]. Conversely, attempts to extrapolate foraging patterns from earth observation data habitat mapping are not based on validated matching of the timing of plant flowering with pollinator activity [9,14]. Such approaches fail to capture the diversity of local floral resources resulting from management, regional species pools and historical legacy effects [14]. Addressing this evidence gap provides opportunities to understand the impacts of existing and policy driven changes in land use and management that can inform spatially and temporally explicit mitigation measures to support pollinators [12].

The European honeybee (*Apis mellifera* L.) has the potential to act as a model biomonitoring system for understanding impacts of local, national and long-term environmental drivers impacting on floral resources critical for wild pollinators in general. As a group, they have been widely used to monitor environmental change and pollutant exposure [15–19]. Their application for these purposes extends from advantages, including: 1) honeybees are managed as livestock, such that disease control and supplementary feeding means that they are kept in almost every environment with c. 288,000 hives in the UK. As a result, a comprehensive geographical cover exists that in practice is not seen for wild bee species; 2) beekeepers represent an

engaged community willing to participate in citizen scientist projects with a demonstrated track record of providing stored hive products like honey for scientific studies [19,20]; 3) there are limited ethical considerations for the collection of stored products from hives compared to comparable sampling from wild bee species. This is particularly true where sampling involves killing wild pollinators or placing artificial colonies, e.g., commercial bumblebee colonies, in natural conditions; 4) honeybees provide insight into wider pollinator resource utilisation. While each species of pollinator differs in their foraging preferences (e.g., oligolectic or polylectic species), as a generalist forager honeybees provide information on the range of plant species likely to be in flower and so available to many species [21,22]; 5) honeybees integrate information on foraging resources over large areas, with mean foraging distances on average of 1.4 km from hives [23].

Here we report on the UK National Honey Monitoring Scheme (NHMS) which establishes a large-scale biomonitoring platform that provides long-term monitoring of bee resource utilisation as well as an archive of samples to be used to assess other environmental drivers on bees, including pesticides [24,25]. This scheme does not itself promote beekeeping, which has implications for competition with wild pollinators [26], but takes advantage of an existing network of some 30,000 beekeepers in the UK. While other large scale biomonitoring using honeybees has been implemented, we detail the establishment and maintenance procedures underpinning such large-scale programs over multiple years [27]. We also describe how environmental DNA (eDNA) based approach for plant identification have been used to analyse DNA derived from honey (majority from pollen) to identify foraging plants linked to explicit spatial, temporal and other beekeeper provided metadata [28–32]. This paper provides an overview of data generated to-date and focuses on the value of citizen scientist beekeepers and honeybees as a cost-effective approach for biomonitoring at national scales.

## Materials & methods

### Establishment of the scheme

The scheme was initiated by the UK Centre for Ecology & Hydrology (UKCEH) and was officially launched on 1st July 2018 after consultation with the UK's national apicultural bodies for amateur (the British Beekeepers Association – BBKA) and professional (Bee Farmers Association – BFA) beekeepers. This initial consultation phase was used to identify limiting factors to uptake of the scheme, key approaches to publicising and acquiring beekeeper participants, expectations from participants of data dissemination, as well as approaches to limit participation fatigue. Stakeholders defined the scope of additional metadata that may be collected from beekeepers including information on hive productivity, management diseases and their treatment. Stakeholder engagement prompted the decision to focus on honey stored in combs as opposed to pollen stored as beebread. It was advised that largescale collection of pollen either directly from combs or using pollen traps would be less appealing to beekeepers and significantly reduce participation. Stakeholder consultation was also used to design user-friendly web-based portals as well as the type and frequency result dissemination to beekeepers (S1-S2 Figs). This included proposals for email updates, member web portals for accessing results and newsletters.

### Recruitment of beekeepers

Advertising the scheme was achieved though the beekeeper association newsletters, industry targeted magazine articles, promotion at apicultural conventions, a website (S1 Fig) and Twitter/X account. The website was also used as the portal for members to participate and order sampling packs. Outreach to the beekeeping community was integrated to maintaining interest in the scheme though presentations at national or regional beekeeping shows or for local beekeeping societies. Recruitment of beekeepers into the scheme started on the 1.3.2018 and has continued since then in this ongoing scheme. Continued recruitment has been required to improve the geographical spread of samples as well as to account for participants who have left the scheme. Participation required the creation of an online password protected account though the scheme submission portal within which electronic confirmation of consent for participation in data collection is obtained. The scheme does not include minors.

## Honey sample collection by beekeepers

Participation in the scheme begins when beekeepers create an account on the online portal – accessed *via* the NHMS website – and in doing so provide the location of the hive they will sample (Fig 1; S2 Fig). This indicates the sample pack request process. Sample packs are posted to the beekeeper containing three sterile 30 ml sample tubes and sampling instructions (S3 Fig). Beekeepers are asked to fill each of the three tubes with honey collected directly from recently laid down comb (typically produced in the prior 1–2 weeks) and are supported in doing this with links to professional beekeeper online tutorial demonstrations. We do not use commercially spun honey. Sample tubes are labelled with the collection date and returned to the scheme though a pre-paid postal system. The bee-keeper receives an email update each time the status of their sample changes (e.g., sample pack requests or successfully returned) or when information is added to the online portal. Once a beekeeper has created an account and requested a sampling pack, they were requested to provide additional metadata relating to beekeeper experience, number of hives in their apiary, as well as symptomatic diseases found within the hive from which samples were taken (S1 Table). Provision of this additional metadata was on a voluntary basis and not all beekeepers provided this information. Further details of sample pack requests, initial sample processing can be found in Supplementary Methods.

## Using eDNA barcoding to identify forage plants

While honey is ostensibly nectar in origin, it contains suspended pollen grains that provide information on honeybee diets. Pollen is ideally suited to this process as it is both ubiquitous and robust to degradation, with DNA markers that differ between species but are relatively conserved within species [33]. The identification of such pollen grains has a commercial use in terms of validating the origin of honey sold for human consumption, e.g., for monofloral honeys like Manuka. Full details of the protocol for detecting plant taxa in NHMS honey samples are published in Oliver, Newbold (34). In summary, an initial small sample from the tube is extracted to test moisture content using a handheld refractometer (S4 Fig) with samples deviating from the expected 80% sugar being noted as being at risk from fermentation or microbial growth. Then 15 g of honey is transferred into a 50 ml tube, which is filled to 50 ml with molecular grade water (Corning, USA) and heated at 56°C for one hour with regular vortexing. Wax is then filtered from the honey-water mixture using a Stomacher® bag (Seward, UK) and the filtrate passed through a sterile 1.2 µm membrane filter (Merck, Germany) using Nalgene reusable bottle top filter assembly. Vacuum suction is applied to the Nalgene filter to draw the honey-water filtrate through the membrane filter such that pollen grains and associated DNA are collected on the membrane filter. This membrane filter is folded into a cryogenic vial (Corning, UK) and stored at -80°C.

DNA is then extracted from filters using a modified protocol for the DNeasy® PowerPlant Pro Kit (Qiagen, Germany) [34]. Between 0.5-2 ng of extracted DNA is amplified using universal plant metabarcoding primers internal transcribed spacer 2 (ITS2) region in combination with dual-index barcodes [35,36], allowing the pooling of 384 honey samples into one sample for sequencing, and optimised for sequencing using Illumina MiSeq v3 chemistry (Illumina, USA). Prior to sequencing, each PCR amplicon is visualised by gel electrophoresis and any sample without a strong band in the region of 450–500 bp is not taken forward to sequencing. Following sequencing, sequences are de-multiplexed to originating samples and processed through the open access HONEYPI pipeline [37]. Data output from the HONEYPI pipeline consists of an identified list of taxa, and number of associated sequences found in each sample. Taxa identified as plants are phylotyped (grouped) to most resolved taxa, usually to species level. To ensure that curation of data is standardised within and between years quality control of the data is implemented. This involves checking numbers of reads in negative controls to ensure no contamination has occurred and checking there is similar sequencing depth between samples and across the dual-indexed primer plates, to ensure primers are performing equally. Sequenced samples are rarefied to several reads that ensures sufficient sequencing depth whilst maintaining sample numbers and excluding negative controls.

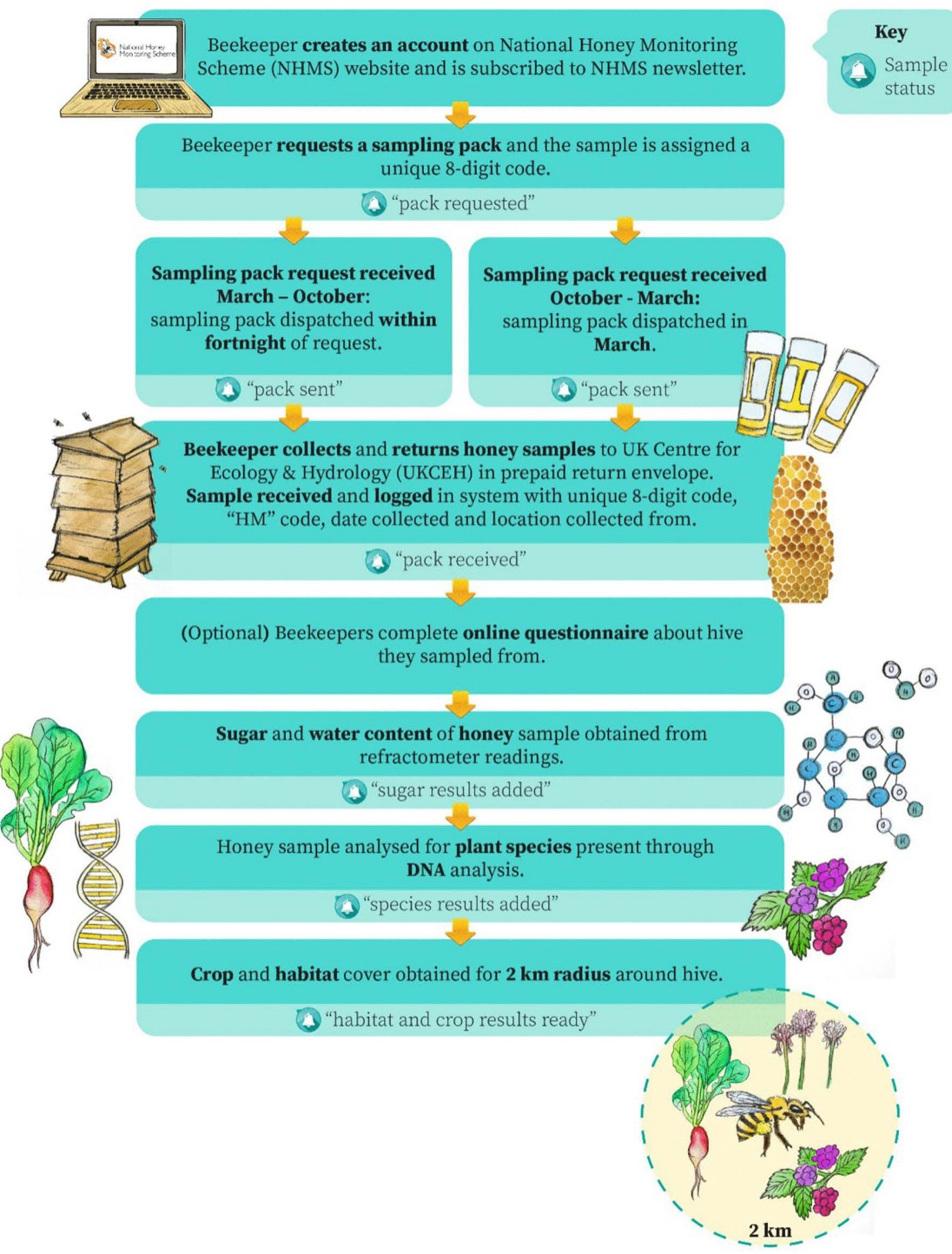

**Fig 1. Flowchart of the process that participating beekeepers and the honey samples go through.** This describes the flow of samples from creating an online account to sample collection and return to UK Centre for Ecology & Hydrology (UKCEH) for analysis..

## Landcover habitat data associated with individual hives

Local habitat data within a 2 km radius of georeferenced sampled hives was extracted each year from satellite-derived land cover and cropping data products (S2 Table). This information is both used as a covariate for potential analyses as well as being provided in feedback to the beekeepers with information on what their bees have been foraging on. The percentage cover of land uses within 2 km of hives is derived from the UKCEH Land Cover Maps (LCM) which currently detail 21 land cover classes for Great Britain and Northern Ireland including arable and horticulture, different classes of grassland, woodland, wetland, and upland systems (S5 Tabel) [38]. We used the finest scale raster datasets available: 25m for all but LCM 2021, when the 10m dataset was released. Due to the timings of LCM publication versus data extraction, earlier data extractions used LCM 2015 while later extractions the previous year's LCM (in practice this delay mirrors processing times for honey eDNA analysis). In addition, the percentage cover of agricultural crops is derived from the UKCEH Land Cover® *plus*: Crops (LC+ Crops) are vector datasets detailing annual crop classes per arable field parcel [39]. Crop classes include important flowering crops, like oilseed rape, field beans and floral grass leys.

## Dissemination of findings to participants

Dissemination of results to participant beekeepers operated at three main levels and follows strict guidelines and precautions to comply with UK General Data Protection Regulations (Supplementary Methods). The first level of data dissemination is at the individual beekeeper level through information provided *via* their online user account. Beekeepers are sent the plant species detected in the honey sample they collected. These data are visible to beekeepers showing the 15 most abundant plant taxa in their honey samples (S5 Fig). In addition to this, information is provided on crops and habitats radius surrounding the hive to a 2 km radius (S6 Fig). The second level of data dissemination is targeted at members of the scheme and provide summary annual statistics through a newsletter. This is intended to maintain scheme interest and help create a community of members to maintain fidelity to the scheme over time. Over time the community has developed independently, with beekeeper forums expressing considerable interest and discussion when results are released. The final level of dissemination to participants is though targeted stakeholder engagement talks provided to both local, regional and national beekeeper organisation.

# Results

## Participation in the scheme

In the scheme's launch year, 2018, 251 beekeepers requested a sampling pack (S3 Table) with this rapidly increasing to a scheme peak of 1,864 in 2022 (Fig 2). As of 2023 the scheme significantly reduced the number of sampling packs provided to beekeepers reflecting the high costs associated with eDNA barcoding to 384 processed samples. However, 1,026 pack requests were still received in 2023 although this reduced to 767 by 2024. Although attempts though scheme promotion have been implemented to recruit more beekeepers from the North of England and Scotland, the East of England, Southeast of England, London and Southwest of England typically composed on average between 55–62% of samples (S4 Table). In part, this bias is believed to reflects regional population densities in the UK which increase the likelihood of hobbyist beekeepers being present in those areas. Efforts to promote increased national coverage meant that from 2023, 49% of samples were from the East of England, Southeast of England, London and Southwest of England, with this falling again to 45% in 2024. Although samples from 2018 were predominantly from July and August reflecting the late start date of the scheme, from 2019 onwards samples were provided from April, with numbers peaking in August whereafter they dropped off. Not all sampling packs sent to beekeepers were returned, however return rates are typically between 70% to 79% (S3 & S4 Tables).

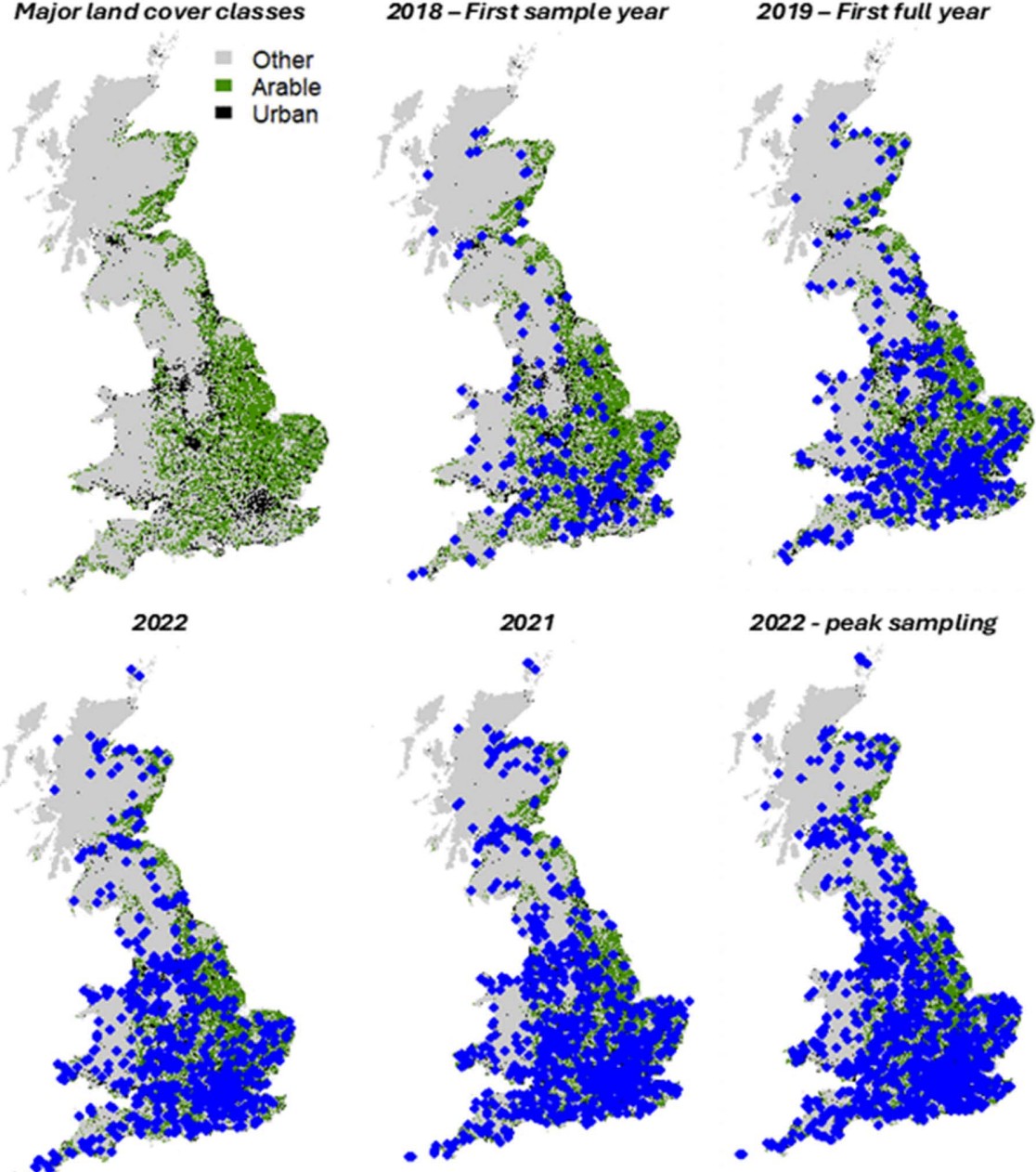

**Fig 2. Location of hives providing honey samples from the National Honey Monitoring Scheme.** Starting with the initiation year in 2018 and including the peak sample collection year of 2022. From 2023 the size of the scheme was reduced reflecting processing costs for eDNA pollen grain analysis.

## Honey eDNA analysis

At the time of writing, honey data from the ITS2 region plant metabarcoding primers had been sequenced and interpreted though the HONEYPI bioinformatics pipeline for years 2018–2022. Processing of following years was ongoing. From 2018 to 2022, plant eDNA sequences were successfully produced for 95–98% of the returned honey samples (S3 Table).

The 2–5% of returned samples for which plant DNA data was not produced failed to produce a strong band upon visualisation by gel electrophoresis, suggesting the DNA extraction and/or PCR amplification step of the Honey DNA analysis pipeline was unsuccessful. In 2020 and 2021, honey DNA data was produced for 70–74% of the returned samples, the remainder being archived. This reflects a combination of the resources available to undertake these analyses as well as the number of samples that can be included in a sequencing run ($n = 384$). From 2018 to 2019 all returned samples fitted into two sequencing runs; however, by 2020 the two sequencing runs fitted 775 of the 1113 (70%) returned samples. In 2021, three sequencing runs were financed and were able to fit most of the returned samples, whereas in 2022 continued increase in the popularity of the scheme meant that the three sequencing runs fitted 74% of the returned samples.

From the honey sample pollen eDNA was resolved to 804 species level classification, 101 generic and 16 family or lower taxonomic resolution though the HONEYPI informatics pipeline. Species include wild plants as well as both ornamental garden and invasive plant species. Each sample is spatially and temporally resolved providing opportunities for understanding long term and large-scale impacts of land use change, management and other external environmental drivers like climate. A detailed breakdown of the top 40 plant genera detected in samples can be found in S7 Fig. Most samples originate from between May and September, during which time five dominant plant genera make up the diets of honeybees. These are the *Brassica* spp. (predominantly the crop oil seed rape but also cruciferous vegetables), *Trifolium* spp. (*clovers,* including white clover common in improved grassland systems), *Impatiens* (sp. *glandulifera*, also known as the highly invasive Himalayan balsam), *Rubus* (brambles) and *Myosotis* spp. (forget-me-not). These five plant genera show strong latitudinal clines across four geographical bands from Scotland to Southern England (Fig 3). There are also clear temporal patterns in foraging resource utilisation with dominance of *Brassica* spp. in May and June samples when the crop oilseed rape flowers and a rise of *Rubus* spp. in July to September when most bramble species flower. *Rubus* spp. is much more dominant in the South and Midlands than in the North or Scotland. By contrast, *Impatiens* – which represents the invasive species Himalayan balsam – is much more dominant in the North than in other geographical bands. Most plant genera detected from DNA in honey in the North, Midlands and South are represented by these five genera, whereas Scotland has a much greater diversity of other species on which the honeybees forage (Fig 3).

## Landscape context of sampled hives

Honeybee hives from which samples were analysed for foraging plant eDNA were located in a wide range of different habitats (Fig 4). On average 54% of land use surrounding within 2 km of hives was composed of intensively managed agriculture, such as arable, horticulture or improved grasslands, while 25% of surrounding land use was either urban or suburban. Agricultural cropping practices within a given year surrounding hives can also be defined (S5 Table) providing opportunities for understanding both dependencies on flowering crops and with this potential exposure risks to intensive management practices, such as pesticide use.

## Recruitment and retention of participant beekeepers

Key to recruitment of the scheme in the early years was ongoing interaction between the scheme researchers and the beekeeping community, including e-mail updates, newsletters and presentations for beekeeping societies. Subscription to the NHMS newsletter is used as an indicator of interest in the scheme. As of June 2025, the NHMS newsletter had a total 4,950 individuals on its mailing list who had created an account on the website from 2018-2025, of which 4,649 (94%) were still subscribed. This means that 301 (6%) individuals opted to unsubscribe from receiving the newsletter at some point after creating an account. Mailchimp communication software has recorded an average open rate of newsletters as 61.5%. As such, with current subscription rates ~4,370 individuals open the newsletter.

## Meta-data provided by beekeepers

The percentage of beekeepers who responded to each of the twelve optional questions in the online questionnaire varied between years, with between 34% and 56% beekeepers responding to the questions (S8 Fig). The most reported issues

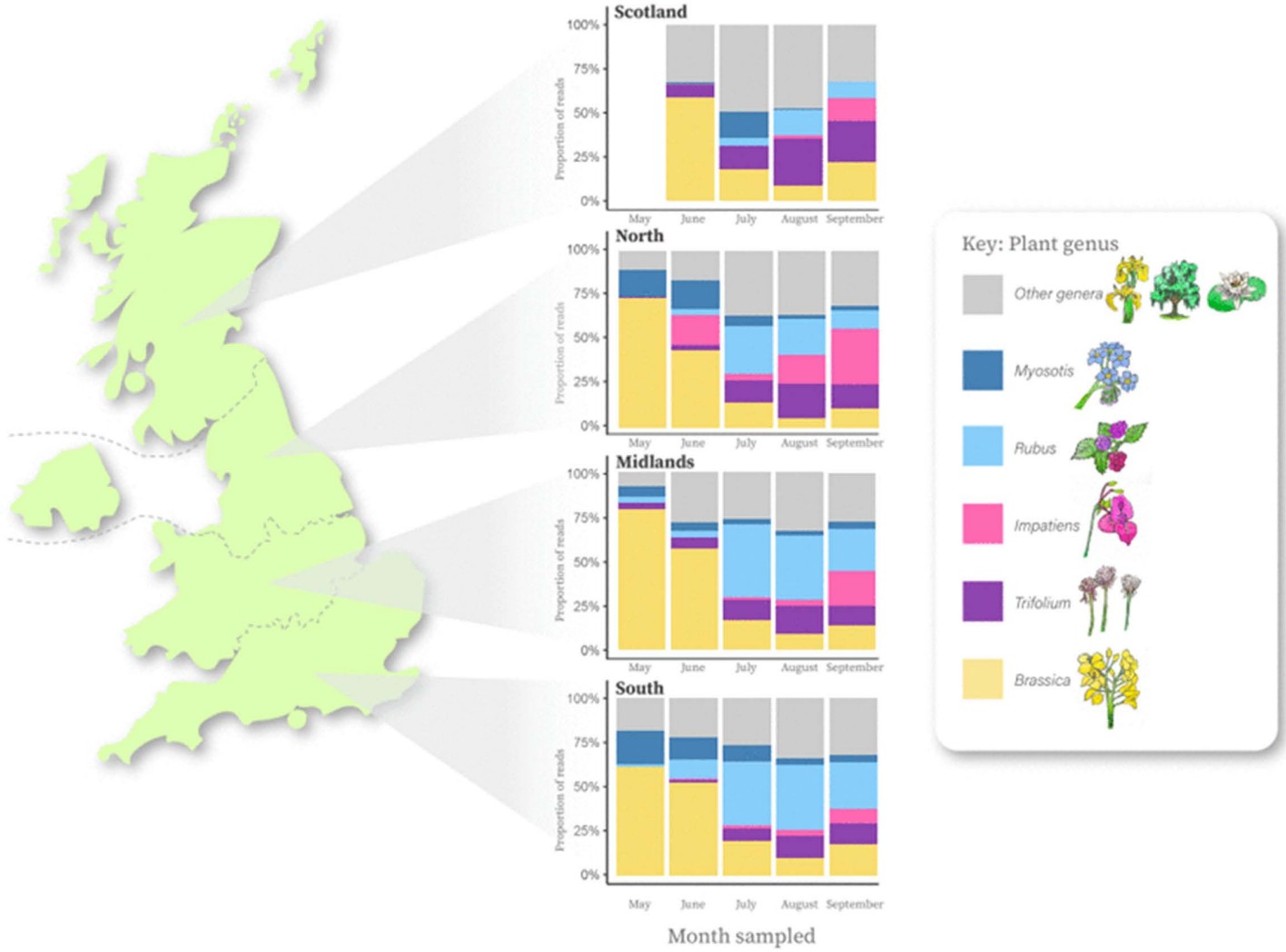

**Fig 3. Distributions of five plant genera (*Brassica, Trifolium, Impatiens, Rubus* and *Myosotis*) detected *via* plant DNA in honey collected from four UK geographical bands.** "Scotland" is comprised of Scotland alone. "North" is comprised of International Territory Level regions for Northwest England, Northeast England, Yorkshire and the Humber, Isle of Man and Northern Ireland. "Midlands" is comprised of Wales, West Midlands, East Midlands and East of England. "South" is comprised of Southeast England, Southwest England and London.

linked to the meta data were failed or failing queen(s), predation by wasps and drone-laying, followed by reports of symptomatic diseases including varroa infestation, as well as chalkbrood, sacbrood, nosema, European foulbrood (EFB) and chronic bee paralysis virus (CBPV).

## Discussion

This paper was intended to show the viability of using honeybees as biomonitors integrating information on resource utilisation of wild, crop and ornamental garden plants at landscape scales, as well highlight other opportunities that such a network may pose for environmental change monitoring [16,18,27,40]. We describe the establishment and running of the National Honey Monitoring Scheme over its first five years and show how this has been operationalised to quantify pollinator foraging resources. This provides underlying data opportunities to understand how these are promoted or limited by

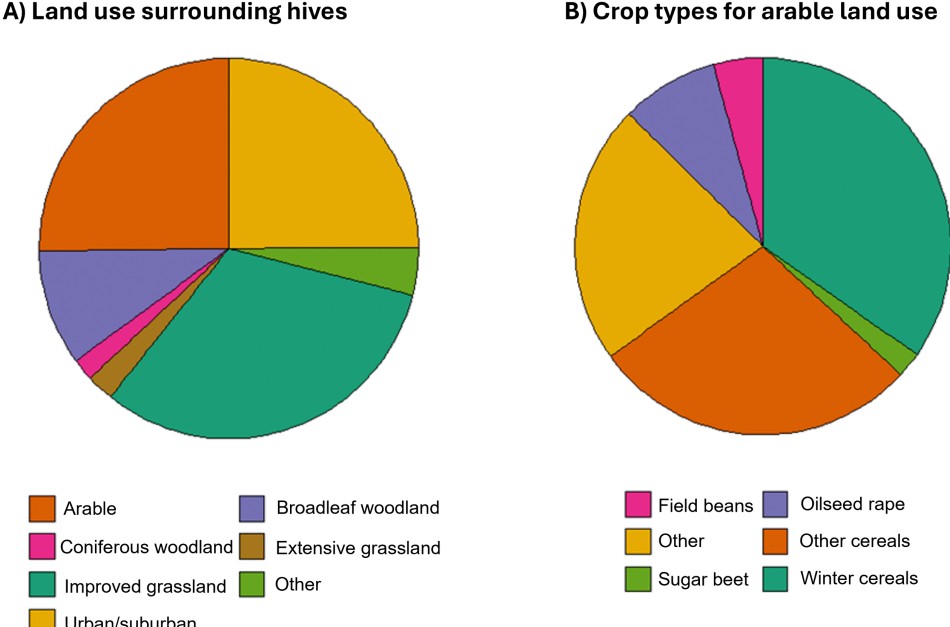

**A) Land use surrounding hives**

**B) Crop types for arable land use**

Legend A:
- Arable
- Coniferous woodland
- Improved grassland
- Urban/suburban
- Broadleaf woodland
- Extensive grassland
- Other

Legend B:
- Field beans
- Other
- Sugar beet
- Oilseed rape
- Other cereals
- Winter cereals

**Fig 4. Surrounding land use of sampled honeybee hives. (A)** Summary average percentage cover of land use types within 2 km of participant hives, and **(B)** percentage of crops within arable land use.

land use, climate, or the actions of government agri-environmental policies. With over 4,950 beekeepers involved between 2018–2024 the scheme has been a citizen science success story and shown rapid growth in participation. At present we believe this to be one of the largest citizen science projects involving eDNA in operation worldwide. By taking a citizen science approach to sample collection the geographical coverage that can be achieved is beyond what could be achieved using conventional researcher-based sampling approaches. Combining collected data with other spatially explicit data sets, including those on landcover data, crop types, pesticide use, or climate, further provide new insights into how environmental pressures impact insect pollinators.

## Opportunities for exploiting the NHMS archives and data

The overview of baseline data showing how such a large-scale monitoring platform can be integrated into molecular eDNA analysis pipelines to quantify actual, as opposed to inferred, resource utilisation by bees. While this provides data specifically on honeybees, it is likely that they act as a model that can be used to infer wider risks for pollinators in general. Resource utilisation by honeybees will not be the same as that of wild bees [21], but as short tongue generalists it is likely they have significant cross over in flowering resources with wild species. This is evidence of competition for floral resources with wild species in some cases [21,22]. Our use of honeybees links to a growing interest in environmental monitoring of pollinator resource requirements and spatial and temporal resource limitations [16,18,41–43]. The NHMS provides an important proof-of-concept for honeybees as an eDNA biomonitoring platform.

The scheme has accumulated 5,789 honey samples in its archive since 2018 and these samples contain a wealth of eDNA. While the current focus to date has been on foraging plant resources determined from pollen grains suspended in the honey, this only touches on the opportunities that archived samples may provide. Perhaps the most obvious is the use of this eDNA to track the emergence and spread of invasive plant species within the UK; with the invasive Himalayan Balsam (*Impatiens glandulifera*) being an example of such a species frequently detected within the honey samples. Such

assessments may also provide insights into how invasive species are integrating into pollinator trophic interactions with wider plant communities [44]. Additionally, the application of alternative molecular techniques to the archived DNA (such as quantitative PCR and metagenomics) has the potential to expand beyond forage distribution to address important questions on how pollinators interact with other multi-kingdom members of the landscape, like fungi, viruses, arthropods and microbes [45–47]. This eDNA may also be used to support detection of honeybee pathogens and parasites within the hive [47], plant pathogens present in the area surrounding the hive [48], or even provide an early warning of Asian Hornets in the vicinity of the hive via detection of airborne eDNA [49].

The archive of honey samples also has value for inferring the scale and temporal variability of environmental pollution effects on honeybees and pollinators in general [27,40]. This may include inferences about how land use (e.g., agricultural, urban and amenity areas) or regulatory policy decisions act to effect exposure [19,20,25,27,40]. From 2019 onwards, pesticide residues have been assessed from honey samples to support the UK Government indicator of exposure and adverse effects of chemicals on wildlife indicator for its 25 Year Environment Plan [24,50]. This is intended to understand long term changes in risk of pesticide exposure from agricultural, domestic and urban amenity use of pesticides and currently uses gas and liquid chromatography–mass spectrometry to quantify, as of 2024, 135 herbicides, fungicides and insecticides [24]. The NHMS therefore provides a cost-effective sampling infrastructure to support post regulatory monitoring of pesticides [27]. Collaborations have been initiated with the wider research community to understand exposure to pharmaceuticals in honey resulting from bioaccumulation in pharmaceutical residues spread as sewage biosolids on arable fields [51], nutritional differences between forage plants, and bee diseases such as Nosema, Foulbrood and Chalkbrood. Detection of diseases may provide new insights into temporal pattens of disease prevalence, both symptomatic and asymptomatic, that may help to manage risks of disease spill over from this managed species to wild species [52–54]. This may provide a pragmatic basis for ground truthing mitigation measures to avoid co-location of hives at high value conservation sites.

## Maximising sustainability of this citizen science scheme

There are multiple reasons for the scheme's success. Firstly, it was setup in consultation with the BBKA, BFA and beekeepers such that those who would be participating in the scheme were involved in its design. This is crucial when designing and implementing any citizen science project to ensure it is accessible and of interest to those you wish to participate but was particularly important for this scheme because collecting honey samples from a hive of ~50,000 bees present challenges that may not be evident to a non-beekeeping researcher. Secondly, the team running the scheme invested substantial time and energy into advertising it through presentations at beekeeping conferences, articles in beekeeping magazines and talks to beekeeping associations. Coupled with word-of-mouth between beekeepers, this engagement resulted in news of the scheme spreading rapidly through the beekeeping community. Thirdly, and this cannot be understated, is the motivation of the beekeepers themselves to take part in the scheme. Likely the greatest motivation for beekeepers is finding out, free-of-charge, what their bees have been foraging on when they receive the results from plant DNA analysis of their honey sample. The low number (4%) of beekeepers who have unsubscribed from receiving the newsletter indicates that interest in the scheme is being maintained, even though only a percentage of these subscribers are submitting samples each year.

## Trends in participation and the future of maintaining the scheme

The increase in participation in the scheme between 2019 (777 sampling pack requests) and 2020 (1,414 sampling pack requests) may have been influenced by the COVID-19 pandemic when lockdown measures resulted in individuals spending more time at home and therefore having greater capacity to register and take part in the scheme. Most honey samples are sent in from southern regions of the UK. It is likely that the distribution of the honey samples we receive reflects the distribution of honeybee hives in the UK, but this cannot be verified as there is no publicly available map of all registered

honeybee hives in the UK. As the scheme asks beekeepers to provide honey samples, it may also be that beekeepers with greater honey yields are more likely to spare honey for testing. Beekeepers in the south of the UK experience greater yields than those in the north due to more favourable weather conditions for honeybee foraging. The NHMS team have made concerted efforts to engage with beekeeping to recruit beekeepers from the North of England, Wales, Northern Ireland and Scotland.

### Biases within the data

As a voluntary participation scheme the NHMS has sampling biases associated with who wishes to participate, and when they decide to do so. There lead to geographical biases, as described already, with sample representation being over-whelmingly from the South of England. Samples provided to the scheme also show strong temporal patterns with the peaks occurring in May/June and August corresponding to honey harvests in Southern England, where most samples have been collected. In the UK this reflects observations of beekeepers that honeybees utilise a first nectar flow from mid-April to end of May and a second nectar flow from mid-June to early August. These distinct nectar flows are reflected in the marked transition of floral resources being foraged throughout the year. The metadata provided by beekeepers, in particular their identification of symptomatic diseases is also prone to potential bias. As for any self-reported data its robust-ness is dependent on individual expertise which will differ and with-it beekeeper ability to attribute symptoms to disease. However, the scheme does capture in meta-data how long the beekeeper has been practicing providing some opportunity to account for experience.

### Conclusions

Environmental monitoring is critical to understand long term changes in environmental quality needed to provide the evidence base to supporting large-scale mitigation measures as well as to assess their efficacy in addressing those risks. However, cost for such monitoring is often highly prohibitive and, in many cases, this means that significant limitations are placed on temporal or spatial replication, or emphasises indirect approaches to understand the impacts of environmental change like earth observation satellite imagery. We have shown how citizen scientist beekeepers engaged though the National Honey Monitoring Scheme can provide ground trothed information on plant species occurrence and its resource utilisation by the generalist pollinator honeybees, integrating information on semi-landscape scales reflecting the large foraging distances around hives of the bees. The scheme also provides an archive of samples suitable for understanding other environmental pressures on honeybees, such as diseases and pesticides, and so provides a resource for under-standing future research questions. Beekeepers represent a highly engaged and scientifically minded community that can delivering cost-effective national scale monitoring to understanding how environmental change impacts pollinator health and success. Given the huge economic value of pollinators in supporting crop and wild plant pollination as well as the socio-economic role of beekeeping specifically this information has multifaceted value. This includes supporting regional and national policy decisions in a reactive way as well as providing an evidence base for conservation non-governmental organisations.

### Supporting information

**S1 Fig. Homepage of the National Honey Monitoring Scheme (NHMS).** This is the website (https://honey-monitoring.ac.uk/) that beekeepers interested in participating in the scheme are directed to for more information, and to create an account and request a sampling pack.
(TIF)

**S2 Fig. Citizen scientist location of hives in web portal.** Upon requesting a sampling pack, beekeepers are asked to provide the location of their hive and how long the hive has been at this location. The hive is located using a postcode

search followed by selection of a 10 m square using OpenStreetMap (OpenStreetMap Foundation, UK) or Ordnance Survey Leisure (Ordnance Survey, UK), which is automatically converted to a ten-digit British National Grid reference This map is using the UK Centre for Ecology & Hydrology (UKCEH) site in Wallingford, Oxfordshire as an example hive location.
(TIF)

**S3 Fig. Letter included in sampling packs.** Details of the National Honey Monitoring Scheme (NHMS) and providing instructions for taking part.
(TIF)

**S4 Fig. Testing honey sample sugar content.** A small amount of the honey designated for pollen analysis is spread onto a handheld refractometer, which is held up to the light to be read through the eyepiece. Brix% refers to the sugar content of the honey and water% refers to the water content of the honey. In this example, the sugar content would be recorded as 82.5% and the water content as 15.5%.
(TIF)

**S5 Fig. Web portal-based reporting of plant species to beekeepers.** Identities of plant species detected in each honey sample are uploaded to the online system. Beekeepers are notified that results are available and are able to log into their account to view this report, which will be specific to their honey sample.
(TIF)

**S6 Fig. Reporting land use surrounding hives to beekeepers though the web-based portal.** Crop and habitat types present in the 2 km surrounding the hive, according to UKCEH Land Cover Maps, are uploaded to the online system. Beekeepers are notified that results are available and can log into their account to view these pie-charts, which will be specific to their hive.
(TIF)

**S7 Fig. Plots showing breakdown of top 40 plant genera detected in honey samples.** This is split by month (May to September) and by UK geographical band (Scotland, North, Midlands, South). "Scotland" is comprised of Scotland alone. "North" is comprised of NUTS regions for Northwest England, Northeast England, Yorkshire and the Humber, Isle of Man and Northern Ireland. "Midlands" is comprised of Wales, West Midlands, East Midlands and East of England. "South" is comprised of Southeast England, Southwest England and London.
(TIF)

**S8 Fig. Percentage of beekeepers who answered the optional questions.** Each question was linked to their online account once they had requested a sampling pack in 2018−24.
(TIF)

**S1 Table. Questions asked of beekeepers to provide additional meta-data.** Optional questions relating to honey yield and health of the hive sampled from for the scheme that participating beekeepers can access from their online account.
(PDF)

**S2 Table. Land cover and cropping data sources.** These are defined based on the sample year and the land use type.
(PDF)

**S3 Table. Overview of honey sample collections.** Information on sample returns and data uploads for 2018–2024. Hyphens represent data that was still being generated, at the time of writing.
(PDF)

**S4 Table. Breakdown of samples sent in from NTL regions.** Nomenclature of International Territory Level (ITL) region within the UK, ordered from north to south, from 2018−24. Number in parentheses indicates percentage of returned samples for that year.
(PDF)

**S5 Table. Breakdown of crop and landcover types.** Breakdown of crop and landcover types in 2 km surrounding sampled hives. At the time of writing, habitat data was still being generated for 2023 and 2024 samples. Values given as percentages.
(PDF)

**S1 File. Supplementary methods.** Details of the implementation of the citizen scientist scheme for sampling honey.
(PDF)

## Acknowledgments

The scheme would like to thank all its members who have provided honey samples since 2018 and continue to show enthusiasm for its results. Thanks to the BBKA and BFA for ongoing support and early stakeholder engagement. Special thanks to Ged Marshal for advice and help in producing training videos.

## Author contributions

**Conceptualization:** Lindsay Newbold, Anna Oliver, Daniel S. Read, David Roy, Richard Pywell.

**Data curation:** Jennifer M. G. Shelton, Lindsay Newbold, Emily V. Upcott.

**Formal analysis:** Jennifer M. G. Shelton, Lindsay Newbold, Anna Oliver, Mike Brown, Emily V. Upcott, David Roy.

**Funding acquisition:** Richard Pywell.

**Investigation:** Jennifer M. G. Shelton, Lindsay Newbold, Anna Oliver, Joanna Savage, Eleanor Grove, Manasa Suresh, Michael Bennett, Hyun Gweon Soon.

**Methodology:** Jennifer M. G. Shelton, Lindsay Newbold, Anna Oliver, Emily V. Upcott, Daniel S. Read, David Roy, Hyun Gweon Soon, Richard Pywell.

**Project administration:** Jennifer M. G. Shelton, Lindsay Newbold, Anna Oliver, Joanna Savage, Ujala Syed, Mike Brown, Biren Rathod, Jim Bacon, Katharine Turvey, David Roy, Richard Pywell.

**Software:** Mike Brown, Biren Rathod, Jim Bacon, Hyun Gweon Soon.

**Supervision:** Lindsay Newbold.

**Visualization:** Ben A Woodcock, Lauren Cook.

**Writing – original draft:** Jennifer M. G. Shelton, Ben A Woodcock, Lindsay Newbold.

**Writing – review & editing:** Jennifer M. G. Shelton, Ben A Woodcock, Lindsay Newbold, Emily V. Upcott, Daniel S. Read, David Roy, Hyun Gweon Soon, Richard Pywell.

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
