## [Decision Letter · Decision Letter 0]

11 Feb 2026

PONE-D-25-65088Using honeybees for national scale long-term eDNA biomonitoringPLOS One

Dear Dr. Woodcock,

Thank you for submitting your manuscript to PLOS ONE. After careful consideration, we feel that it has merit but does not fully meet PLOS ONE’s publication criteria as it currently stands. Therefore, we invite you to submit a revised version of the manuscript that addresses the points raised during the review process. As you can see, both reviewers appreciated your study but one had some minor suggestions for improvement, which I would like you to consider.

If applicable, we recommend that you deposit your laboratory protocols in protocols.io to enhance the reproducibility of your results. Protocols.io assigns your protocol its own identifier (DOI) so that it can be cited independently in the future. For instructions see: https://journals.plos.org/plosone/s/submission-guidelines#loc-laboratory-protocols. Additionally, PLOS ONE offers an option for publishing peer-reviewed Lab Protocol articles, which describe protocols hosted on protocols.io. Read more information on sharing protocols at . Additionally, PLOS ONE offers an option for publishing peer-reviewed Lab Protocol articles, which describe protocols hosted on protocols.io. Read more information on sharing protocols at https://plos.org/protocols?utm_medium=editorial-email&utm_source=authorletters&utm_campaign=protocols..

We look forward to receiving your revised manuscript.

Kind regards,

Olav Rueppell

Academic Editor

PLOS One

Journal Requirements:

Reviewers' comments:

Reviewer's Responses to Questions

**Comments to the Author**

1. Is the manuscript technically sound, and do the data support the conclusions?

Reviewer #1: Yes

Reviewer #2: Yes

2. Has the statistical analysis been performed appropriately and rigorously? 

Reviewer #1: N/A

Reviewer #2: Yes

3. Have the authors made all data underlying the findings in their manuscript fully available?

Reviewer #1: Yes

Reviewer #2: Yes

4. Is the manuscript presented in an intelligible fashion and written in standard English?

Reviewer #1: Yes

Reviewer #2: Yes

5. Review Comments to the Author

Reviewer #1: The article presents an interesting study on citizen science and the survey of eDNA in the honeycomb. The samples were collected by farmers and hobbyists across the United Kingdom. One thing I greatly appreciate about the article is the description of the process of engaging with citizens across the UK, including the use of websites to create surveys and collect metadata. The article also presents the numbers of samples and the issues surrounding their collection and processing.

The results are generally descriptive and could be considered a bit light. However, they do demonstrate the utility of the DNA results. They were able to relate the honey source to multiple plant types across the UK. This can be quite valuable for understanding the foraging behaviour of honey bees, which are under threat from pollution and other anthropogenic pressures, but also for monitoring the landscape and climate utility.

How could the DNA information be quite useful? Other things like predicting or monitoring plant diseases. It can also be used to monitor pathogens in bees. this process could also be expanded to other insects, which the authors explain in the discussion.

It was a good, informative read. Overall, the paper describes a clever and interesting project with potential future implications. It may not be very scientific or hypothesis-driven, but this information is good to know. It demonstrates the good utility of citizen science and could help further scientific investigations. This paper is good: it demonstrates the effective utility of citizen science, announces that this resource is available, and can serve as a good stepping stone (template) for those seeking to use citizen science or landscape biomonitoring.

Reviewer #2: This manuscript summarizes the results of the UK National Honey Monitoring Scheme, where they recruit volunteer beekeepers to collect and submit honey samples from hives located around the United Kingdom, which they then extract pollen DNA from and use DNA sequencing in order to identify and quantify which plant species the bees were foraging on. They summarize five years of data, and are able to show both geographical and chronological variation in the plants being foraged on by the bees.

As someone who is both a geneticist and an amateur beekeeper, I thought that this was a very fun paper to read. The results seem well supported by their data, and should be of interest to both scientists as well as members of the public. I have no major criticisms of the paper, although I did find a few minor grammatical errors that should be corrected before publication.

Line 57: should be "significant implications FOR understanding"

Line 246: missing word "from" after "honey data"

Line 268: the word "including" should not be italicized

Line 339: should be "evidence OF competition"

Figure 4a: The "Urban/Suburban" box in the legend is cut off on the bottom.

Figure 4b: "Arable" is misspelled in the figure title. "Other Cereals" is misspelled in the figure legend.

6. PLOS authors have the option to publish the peer review history of their article (what does this mean?). If published, this will include your full peer review and any attached files.). If published, this will include your full peer review and any attached files.

.

Reviewer #1: No

Reviewer #2: No

---

## [Author Response · Author response to Decision Letter 1]

30 Mar 2026

Editor

Journal Requirements:

Comment: If the reviewer comments include a recommendation to cite specific previously published works, please review and evaluate these publications to determine whether they are relevant and should be cited. There is no requirement to cite these works unless the editor has indicated otherwise. Response: no additional references were suggested.

Comment: Please review your reference list to ensure that it is complete and correct. If you have cited papers that have been retracted, please include the rationale for doing so in the manuscript text, or remove these references and replace them with relevant current references. Any changes to the reference list should be mentioned in the rebuttal letter that accompanies your revised manuscript. If you need to cite a retracted article, indicate the article’s retracted status in the References list and also include a citation and full reference for the retraction notice.

Reviewer #1

Comment: The article presents an interesting study on citizen science and the survey of eDNA in the honeycomb. The samples were collected by farmers and hobbyists across the United Kingdom. One thing I greatly appreciate about the article is the description of the process of engaging with citizens across the UK, including the use of websites to create surveys and collect metadata. The article also presents the numbers of samples and the issues surrounding their collection and processing. Response: Thankyou for these comments. This sis a large and often hard to run citizen science program. As you say we were hoping not only to provide an overview of the scope of the data being created, but also to support the wider community in processes and methods for setting up this kind of project.

Comment: The results are generally descriptive and could be considered a bit light. However, they do demonstrate the utility of the DNA results. They were able to relate the honey source to multiple plant types across the UK. This can be quite valuable for understanding the foraging behaviour of honey bees, which are under threat from pollution and other anthropogenic pressures, but also for monitoring the landscape and climate utility. Response: Thank you.

Comment: How could the DNA information be quite useful? Other things like predicting or monitoring plant diseases. It can also be used to monitor pathogens in bees. this process could also be expanded to other insects, which the authors explain in the discussion. Response: thank you. Indeed very recent work we have done is starting to show how this honey national archive can be used with outputs coming out since submission showing how the honey collected under this scheme can be used to identify bee diseases, as well as pharmaceutical residues resulting from spreading of sewage sludge on agricultural land.

Bennett, M.J.R., Newbold, L.K., Busi, S.B., Pywell, R., Tipper, H., Savage, J., Shelton, J., Suresh, M., Grove, E., Gweon, H.S. & Woodcock, B.A. (2026) Molecular detection of honeybee pathogens in honey from a UK citizen science program. Journal of Microbiological Methods, 107451.

Nightingale, J., Woodcock, B.A., Garazade, N., Pywell, R.F. & Carter, L., .J. (2026) Presence of Emerging Contaminants in UK Honey - Human Pharmaceuticals a Concern for Honeybees? Journal of Agricultural and Food Chemistry, https://doi.org/10.1021/acs.jafc.5c10414.

Comment: It was a good, informative read. Overall, the paper describes a clever and interesting project with potential future implications. It may not be very scientific or hypothesis-driven, but this information is good to know. It demonstrates the good utility of citizen science and could help further scientific investigations. This paper is good: it demonstrates the effective utility of citizen science, announces that this resource is available, and can serve as a good stepping stone (template) for those seeking to use citizen science or landscape biomonitoring.

Response: thank you.

Reviewer #2:

Comment: This manuscript summarizes the results of the UK National Honey Monitoring Scheme, where they recruit volunteer beekeepers to collect and submit honey samples from hives located around the United Kingdom, which they then extract pollen DNA from and use DNA sequencing in order to identify and quantify which plant species the bees were foraging on. They summarize five years of data, and are able to show both geographical and chronological variation in the plants being foraged on by the bees. Response: Thanks you for this summary.

Comment: As someone who is both a geneticist and an amateur beekeeper, I thought that this was a very fun paper to read. The results seem well supported by their data, and should be of interest to both scientists as well as members of the public. I have no major criticisms of the paper, although I did find a few minor grammatical errors that should be corrected before publication. Response: thank you for these kind comments. This paper is hoped to both publicise the project and set the scene for future work utilising this data.

Comment: Line 57: should be "significant implications FOR understanding". Response: changed as suggested.

Comment: Line 246: missing word "from" after "honey data". Response: changed as suggested.

Comment: Line 268: the word "including" should not be italicized. Response: changed as suggested.

Comment: Line 339: should be "evidence OF competition". Response: changed as suggested.

Comment: Figure 4a: The "Urban/Suburban" box in the legend is cut off on the bottom. Response: Corrected as suggested.

Comment: Figure 4b: "Arable" is misspelled in the figure title. "Other Cereals" is misspelled in the figure legend. Response: Corrected as suggested.

---

## [Editor Report · Decision Letter 1]

3 Apr 2026

Using honeybees for national scale long-term eDNA biomonitoring

PONE-D-25-65088R1

Dear Dr. Woodcock,

We’re pleased to inform you that your manuscript has been judged scientifically suitable for publication and will be formally accepted for publication once it meets all outstanding technical requirements.

An invoice will be generated when your article is formally accepted. Please note, if your institution has a publishing partnership with PLOS and your article meets the relevant criteria, all or part of your publication costs will be covered. Please make sure your user information is up-to-date by logging into Editorial Manager at Editorial Manager® and clicking the ‘Update My Information' link at the top of the page. For questions related to billing, please contact  and clicking the ‘Update My Information' link at the top of the page. For questions related to billing, please contact billing support..

Kind regards,

Olav Rueppell

Academic Editor

PLOS One
---

## [Editor Report · Acceptance letter]

PONE-D-25-65088R1

PLOS One

Dear Dr. Woodcock,

I'm pleased to inform you that your manuscript has been deemed suitable for publication in PLOS One. Congratulations! Your manuscript is now being handed over to our production team.

Kind regards,

on behalf of

Dr. Olav Rueppell

Academic Editor

PLOS One